# The Heterogeneous Effects of Participation in Shadow Education on Mental Health of High School Students in Taiwan

**DOI:** 10.3390/ijerph18031222

**Published:** 2021-01-29

**Authors:** I-Chien Chen, Ping-Yin Kuan

**Affiliations:** 1College of Education & CREATE for STEM Institute, Michigan State University, East Lansing, MI 48824, USA; 2Department of Sociology & International Doctoral Program in Asia-Pacific Studies, National Chengchi University, Taipei 106011, Taiwan; soci1005@nccu.edu.tw

**Keywords:** private supplementary education, shadow education, depression, entrance examination, secondary education

## Abstract

The effect of shadow education or private supplementary education (PSE) on school achievement has been prolifically studied, but its impact on well-being remains understudied. This study examines the heterogeneous effect of PSE participation on school achievement and depression symptoms among high schoolers in Taiwan. The study uses panel data of the Taiwan Upper Secondary Database (TUSD) in the 2014 and 2015 academic years. We join the inverse-probability-of-treatment weighting (IPTW) approach and the seemingly unrelated regression (SUR) model to estimate the effects of PSE participation patterns on two correlated outcomes, comprehensive assessment of high school entrance examination and self-reported depression symptoms in the 11th grade. The latent class analysis identifies five PSE participation patterns: always-taker, early-adopter, dropout, late-adopter, and explorer, to predict the effect of PSE on the scores of entrance examination and later depression symptoms in high school (*n* = 7708, mean age = 15.33). The findings suggest that PSE participation in junior high is positively associated with academic achievement. However, PSE participation also increases depression symptoms, particularly in the case of always-takers. In other words, while always-takers increase their school achievement in transition into high school, their risks of suffering from depression are also higher than their peers.

## 1. Introduction

Private supplementary education (PSE) is a form of shadow education and is commonly called “*buxi*” or cram schooling in Taiwan. PSE offers private, fee-paying supplementary education for academic subjects beyond school walls [1,2,3]. Hence, one of the primary purposes of PSE participation is to obtain high scores at the nationally administered school or college entrance examination, especially in countries where the competition for enrolling into a more prestigious high school or university is fierce [1,4,5,6,7]. Education researchers noted the prevalence of PSE participation in East Asian countries in the 1990s [8,9]. However, PSE participation has become a worldwide phenomenon in the 21st century [3,10].

The prevalence of PSE is certainly not good news, since the burnout or workload outside of school may cause the development of depression symptoms [11], sleep deprivation [12], and stress in school [13,14,15,16,17,18]. According to a statistical report by the Ministry of Education [19] in Taiwan, nearly 50 to 70% of junior high students (age 13–15) participated in PSE, which is high even among East Asian societies [3]. Parents’ expectations and competitive pressure to outperform peers in school results in students carrying weight on their shoulders from middle to high school.

Previous PSE research, in general, has shown the positive effect of PSE on students’ academic achievement [6,7,13,20,21] and entrance examination [2,3,4,8,22,23]. The positive effect of PSE on academic performance may relate to additional learning resources and opportunities outside the school walls. However, intensive PSE participation did not guarantee success in school achievement. Students showed decreased school performance and felt sleepy during the daytime when they spent over 10–12 h per week participating in after-school studying [12,13]. Moreover, not many empirical studies address the potential negative “side effect” of PSE on young people’s mental health. Depression is one of the fastest-growing illnesses among young people and is found to be one of the most common reasons contributing to adolescent suicide [24,25], anxiety and stress [26,27], and school failure among adolescents [28,29]. It is imperative to understand the potential impacts of PSE participation on adolescents’ mental health.

Is there a trade-off between mental health and school achievement? Or is this trade-off conditional on specific PSE participation patterns? There are even fewer studies investigating the relationship between PSE and adolescents’ mental health [11,30]. In light of the rising popularity of PSE participation worldwide, an empirical study of the effects of various PSE participation patterns on mental health and academic achievement in Taiwan, where the prevalence and intensity of PSE participation are high, would help to fill the gap in knowledge on the short- and long-term impacts of PSE involvement.

The goal of this study is threefold. First, we identify the typical patterns of students’ PSE participation from 7th–9th grade in Taiwan. Second, we assess whether PSE participants outperform non-participants in their entrance examination across different types of PSE involvement. Third, we also analyze PSE patterns’ effects on students’ depression symptoms in later adolescence (11th grade), controlling for early school achievement and mental health. In short, we focus not only on the various patterns of PSE participation but also on the impacts of these patterns on the correlated outcomes of school achievement and depression.

## 2. Materials and Methods

### 2.1. Data and Procedure

The present study uses recently released data gathered by the Taiwan Upper Secondary Database (TUSD) during the 2014 academic year. The TUSD conducted an annual census of first-year senior high school students, with the Ministry of Education’s support, since 2001 when students just enrolled in school. Since 2002, the TUSD also conducted a follow-up census every year when students were in the second semester of their second high-school year. A significant portion of information gathered from the first-year student’s census is about students’ junior high schooling experiences. In 2014, the census gathered extensive information about students’ PSE participation and cram schooling experiences. The response rate of the 2014 census nears 91%. Recently, the TUSD released a randomly selected sample of the 2014 census and its 2015 follow-up census. The sample size is 18,450, which is about 8% of the census data. The present study focuses on those who attended public junior high school and excludes vocational high, private high, and five-year junior college students. We generate missing flags for several covariates, such as parent education, parent occupation, and family structure, to control the effects of missing cases. The final analytical sample size is 7708.

### 2.2. Variables and Measurement

#### 2.2.1. Dependent Variables

The first outcome of interest is self-reported scores from the high school entrance examination. The examination score is the summed score of six test subjects, including Chinese, English, Mathematics, Sciences, Social Studies, and Writing. It ranges between 1 and 37, with a mean of 20.55 and a standard deviation of 7.72. Even though the outcome is self-reported examination scores, the percentile distribution of the self-reported scores correlates strongly with the distribution reported by the Research Center for Psychological and Educational Testing, which is responsible for designing the examination. Hence, the student’s self-reported entrance examination score is a valid measurement of the school achievement outcome in high school transition.

The second outcome of interest is depression symptoms. Depression symptoms are common during adolescence. Symptoms tend to affect students negatively in terms of how they feel and how they act in school. The TUSD survey asked 11th graders about their mental health in the past three months, including (1) I feel like a failure in my life; (2) I feel disappointed in myself; (3) I feel happy; (4) I feel too tired to do anything; (5) I have trouble keeping my mind focused. The five questions are on a scale from 1 to 5. The Cronbach’s alpha is 0.83. These items are similar to the Taiwanese Depression Questionnaire [31,32]. We use the 2-parameter logistic (2PL) item response theory (IRT) model to generate an IRT score for polytomous response data.

#### 2.2.2. Key Independent Variables

To identify individual differences in PSE participation experiences between 7th and 9th grade, we code the participation in each of the six semesters dichotomously and apply the latent class analysis (LCA). Based on the LCA, we identify five PSE participation patterns from 7th to 9th grade. We name these five patterns as always-takers, early adopters, late adopters, dropouts, and explorers. We report the LCA results and class identification index in Appendix A. The decision to choose five patterns as the best fitting solution among alternatives is based on indices of entropy, the Akaike information criterion (AIC), the Bayesian information criterion (BIC), and the Vuong-Lo-Mendell-Rubin likelihood ratio test [33,34]. We also label those who never attended cram schools in junior high as never-takers. The respective count of these six patterns of PSE participation during junior high school are as follows: never-takers, 1888 (24.47%); always-takers, 3659 (47.41%); early adopters, 554 (7.18%); dropout, 534 (7.08%), late adopters, 324 (4.20%) and explorers, 749 (9.71%).

Figure 1 shows the estimated probability of participating in PSE in each semester across five patterns. *Always-takers* are those who attended cram schools for almost all of six semesters. *Early adopters* are students who tended to start attending PSE in the second semester as 7th graders. *Late adopters* are students who began PSE in the 9th grade. *Dropouts* started attending PSE at the beginning of junior high and then drop out of cram schooling in the 8th grade or 9th grade. *Explorers* only participate in cram schooling for one semester, which could occur during any semester.

#### 2.2.3. Covariates

All covariates used to estimate the probabilities of PSE participation patterns and regression analyses are based on the census in the 2014 school year. Demographics and family background variables include the students’ gender, race/ethnicity (Minnan/Mainlander, Aborigine, other), and the number of siblings at home. Student’s characteristics and school environment variables include subject-specific self-efficacy, 9th-grade school ranking, happiness in learning, teacher quality, and classroom climate in junior high.

Subject-specific self-efficacy is measured by the students’ self-report on whether they are confident at five academic subjects (Chinese, English, Mathematics, Sciences, and Social Studies) in junior high. We generate a composite score, which has the Cronbach’s α as 0.86. The 9th-grade school ranking is measured by a self-reported school ranking, ranging from 1–5. Both measures are taken as a proxy for school achievement prior to taking the high school entrance exam. Happiness in learning was measured by asking students whether they felt happy with their learning during junior high, ranging from 1–4; a proxy for mental health in junior high.

Teacher quality or instruction in junior high school was measured by the extent to which students agreed with the following five statements: “How many teachers in your junior high school (1) provide different instructions based on students’ ability; (2) evaluate students’ performance based on other activities (e.g., projects or group discussion); (3) use diverse instructional methods (e.g., projects, group discussions, experiments, experiences or teamwork to accomplish homework); (4) care about your learning (e.g., encourage you or help with problem-solving in learning); (5) teach well or inspire you a lot in learning”. The five questions are on a scale from 1 to 4. The Cronbach’s α is 0.84.

Junior high school climate for learning is measured by the extent to which students agreed with the following statements: “how often the lesson cannot be finished as scheduled”, “how often students skip school”, “student overall learning climate in school”, and “how often bullying behavior happened in school” on a scale from 1 to 4 with higher scores representing a friendly school climate for learning. The Cronbach’s α is 0.70.

The family structure includes five types: (1) intact family—living with both biological parents; (2) step-family—living with the birth mother and the male guardian or the biological father and the female guardian; (3) single-family—residing in the birth-mother-only or the birth-father-only family; and (4) grand-parent guardian, and (5) other guardians/relatives—living with relatives or other non-biological guardians. Parents’ occupation is categorized into six types: (1) farmer or non-technical worker, (2) technical worker, (3) service worker, (4) semi-professional worker, (5) professional worker, and (6) full-time military personnel. The highest parental education is coded into five levels. Either the father’s or mother’s education is categorized into five groups, ranging from “1 = less than high school” to “5 = college-graduated and beyond”. We then use the highest level of education derived from either the mother or father to represent parents’ education. If either of these variables were missing, we substituted either the father’s or mother’s highest education to reduce the missing cases.

### 2.3. Analytical Strategy

This study combines the latent class analysis (LCA), a seemingly unrelated regression (SUR), and the inverse-probability-of-treatment weighting (IPTW) approach to assess the various patterns of PSE participation on students’ entrance examination and depression symptoms in 11th grade. Our analytic strategies are as follows. First, we report the LCA results in Figure 1, which shows the best fitting solution of class identification in five classes (see Appendix A). Second, we provide the descriptive results and the mean difference test compared to the reference group of never-taker in Table 1. Third, we use logistic regression to calculate the inverse probability treatment weighting (IPTW) for the average treatment effect (ATE) and the average treatment effect on the treated (ATT) for each PSE pattern compared to the never-takers [35]. With IPTW weighting, we obtained a pseudo-population where covariates are balanced between the treatment and the control group. We then fit the seemingly unrelated regression model (SUR) with a maximum likelihood estimation of the effects of PSE participation in junior high on two correlated outcomes, high school entrance examination and depressive symptoms in 11th grade (see Table 3 and Table 4). The SUR model is a regression in which two outcomes can be predicted by a set of predictors and accounts for correlations between the residuals, possibly unobserved variables, from the individual regressions. We used the user-written Stata package “mysureg” [36]. Figure 2 shows the research framework and the key variables in the seemingly unrelated regression (SUR) model. We also offer the simplified SURG equation in the following.
(1)$$\left[\begin{array}{c} y_{1}\;\left(Depression\;symptoms\right)\\ y_{2}\left(Entrance\;examination\right) \end{array}\right]=\left[\begin{array}{cc} X_{1} & 0\\ 0 & X_{2} \end{array}\right]\left(\begin{array}{c} \beta_{1}\\ \beta_{2} \end{array}\right)\;+\;\left[\begin{array}{c} \varepsilon_{1}\\ \varepsilon_{2} \end{array}\right]$$

The covariance matrix is
$$\mathrm{Var}\left[\begin{array}{c} \varepsilon_{1}\\ \varepsilon_{2} \end{array}\right]=\;\left(\begin{array}{cc} \sigma_{11} & {}_{}\\ \sigma_{12} & \sigma_{22} \end{array}\right),\;\mathrm{correlation}\;\mathrm{residual}\;(\sigma_{12})$$

## 3. Results

Table 1 presents descriptive statistics across the patterns of PSE participation. Those who are the always-takers during junior high have a clearly advantaged family, school, and educational background and resources. Compared to never-takers, always-takers, early-adopters, and late-adopters tend to have parents with a semi-professional or professional job, at least some college education, middle-class family income with an intact family. The proportion of living in single families and grand-parent families is similar among dropouts, explorers, and never-takers. However, dropouts and explorers tend to have parents with some college education and middle-class family income. Concerning school achievement, PSE participants, regardless of their patterns, have higher entrance examination scores, subject efficacy, and 9th-grade school ranking in junior high than the never-takers. PSE participants also have a smaller number of siblings and a higher family income level than the never-takers.

As for mental health, Table 1 also shows that the always-takers have significantly higher depression symptoms in high school than the never-takers. Students who are early adopters report a lower level of happiness in learning than never-takers. Lastly, all PSE participants experienced a lower school climate level for learning in terms of school characteristics, but only early adopters reported significantly lower teacher quality in junior high. In short, various PSE participation patterns differ in terms of family socioeconomic background, family configuration, and schooling climate experience in junior high. Students with more advantaged family backgrounds tend to participate in PSE for longer or with stability. However, they also tend not to satisfy with their schooling environment. Since these family and school factors are confounders for the observed differences in academic performance and mental health between PSE participation patterns, we need to consider these background differences before we can attribute the differences in outcomes to PSE participation patterns.

To examine PSE participation’s heterogeneous effects on academic achievement and mental health, we used a logistic regression to estimate each PSE pattern’s propensity score with the never-taker as the contrast first. Table 2 shows the results of logistic regression. Consistent with the descriptive findings, being a female, family income, parental education and occupation are all positively associated with the likelihood of being always-takers, early adopters, and late adopters. Those living in a non-intact family, having more siblings, and experiencing a good school climate tend to have a lower likelihood of PSE participation. It is worth noting that living in single or grand-parent families or having more siblings tend to reduce the possibility of being an early adopter and late adopter in PSE. Our logistic regression also confirms that students with an advantaged family background are more likely to participate in PSE. However, different PSE participation patterns also relate to prior school achievement, parents’ education level, occupation, and family structure.

After obtaining the propensity score of PSE participation, we generated the inverse-probability-of-treatment weights (IPTW) for each PSE pattern and conducted a balance check between participants and the never-taker. We use two tests to evaluate the balancing results using the IPTW approach [37,38]. The first was to calculate the covariates’ standardized mean differences between the treatment group (any PSE participation pattern) and the control group (the never-takers). After weighting, the standardized means were between −0.08 and +0.09, markedly smaller than the acceptable range of −0.1 and +0.1 after weighing. The second method uses the model-adjusted difference in the ratio of variances between the treatment and control groups. Rubin (2001) suggests that the range of difference should be between 0.5 and 2.0. The ranges found in our analyses are between 0.8 and 1.2. Most of them are near 1.0. The prior covariates are not statistically significant between PSE participants’ patterns and the never-takers after IPTW weighting.

To estimate PSE participation’s treatment effect, we use the seemingly unrelated regression model (SUR) with maximum likelihood estimation on two possibly correlated outcome variables—entrance examination in the transition into high school and depression symptoms in 11th grade. Table 3 reports the average treatment effect for the treated (ATT) using the IPTW weighted SUR model. For example, the ATT estimated for the always-takers group refers to the difference in the average effect of PSE participation for students who participated in PSE almost all six semesters during junior high school as opposed to the hypothetical (counterfactual) situation if they were never-takers. The findings show some common trends across PSE patterns and suggest that always-takers and dropouts increase their depression symptoms on average by 0.067 (*p* < 0.05) and 0.089 (*p* < 0.05) IRT score. However, they also increase their overall score of entrance examination by 1.942 (*p* < 0.001) and 0.767 (*p* < 0.001). Additionally, early adopters, late adopters, and explorers also increase their overall score of entrance examination by 1.590 (*p* < 0.001), 1.716 (*p* < 0.001), and 1.223 (*p* < 0.001), but they have a similar level of depression symptoms as never-takers in their 11th grade. At the bottom panel of Table 3, the correlation between the residuals of the two outcomes is significant at *p* < 0.01 for always-takers, dropouts, and explorers. In other words, there could be some other unobserved factors linking the two outcomes.

As for the effects of covariates, students had lower depression symptoms in the 11th grade when they were happy in learning in junior high, living in a higher family income household, felt confident in their subject-specific learning, and had a supportive school climate. Students’ parents with a 4-year university degree or semi-professional job increase their depression symptoms. Concerning school achievement, students improved their scores of entrance examination when they had a higher subject-specific efficacy, 9th-grade school academic ranking, and with parents having at least a high school education. Non-intact family structure and a larger number of siblings tend to reduce students’ scores in the entrance examination.

Table 4 shows the results of the average treatment effect of SUR with IPTW weighting. ATE estimates the average treatment effect for each PSE pattern if all students participated in PSE. Specifically, if all students are always-takers, their depression symptoms IRT score would increase by 0.068 and their overall score of entrance examination would increase by 2.071 at the same time. The results are similar to those found for the treated (Table 3), with two exceptions. First, participating in PSE in junior high correlates with increased 11th graders’ depression symptoms among always-takers, dropouts, and late adopters. Second, the correlation between the two outcomes’ residuals is significant for always-takers, early adopters, dropouts, and explorers. As for covariate effects, we found a similar pattern as we described in Table 3. Therefore, we report only the main results of interest in Table 4.

## 4. Discussion

The current study utilizes a person-centered approach to better understand the association between PSE participation in junior high, high school entrance examination and depression symptoms in the 11th grade. It uses a random sample of student census data of secondary education in Taiwan. The analyses combine LCA, SUR, and IPTW to identify common PSE participation patterns, assess the predictors correlated with them, and then estimate the causal effects of various PSE participation on school achievement and mental health in high school.

This study extends our knowledge by demonstrating heterogeneous PSE participation patterns that capture students’ workload and time spent on PSE after school. The present study contributes to understanding the potential trade-off between school achievement and mental health by identifying various long-term PSE participation patterns in junior high based on students’ self-reported experiences. As would be expected from the previous literature, PSE participation tends to increase students’ school achievement [2,4,6,7,20,21,22]. However, previous work has overlooked the potential “side effect” of PSE on young people’s mental health. More importantly, little research has offered insight into the patterns through which PSE participation may trigger health-related consequences for young people. The use of the person-centered approach in the current study extends our previous knowledge by providing a better breakdown of which individual pathways may experience mental health consequences, as a function of PSE participation and high school entrance examination.

The main result relates to the finding that always-takers and dropouts increase their score on the entrance examination but at the cost of increasing their depressive symptoms in the 11th grade. We did not find the same trade-off between school achievement and mental health among early adopters, late adopters, or explorers. In line with our expectations from the early literature [11,12,13,30], being always-takers over the three years may help adolescents keep up with their peers in the entrance examination. However, they may also have high risks of depression symptoms in the 11th grade while controlling for previous school achievement and mental health. In contrast, a PSE late-adopters may gain nearly the same positive effect on the entrance examination as always-takers, but with no apparent mental health cost. These findings shed new light on the possible mental health risks of PSE participants and students’ benefits to the high-stakes entrance examination.

For PSE participants, this study finds a significant correlation between two outcomes among always-takers, dropouts, and explorers, which suggests the entrance examination results may be consequential for students of these types of PSE participation in junior high. Considering how popular PSE participation is in junior high in Taiwan, we expect that this risk factor—in terms of the workload for always-takers and the burnout of dropouts—may result in more severe impacts on adolescents’ later mental health. Further research is needed to examine the effects of PSE intensity, duration, and quality on students’ psychological and physical well-being.

There are some possible explanations as to why PSE always-takers and dropouts suffer more in their later depression symptoms. First, being an always-taker of PSE could indicate that students have difficulties in school achievement. Therefore, students or parents may rely on PSE to help when children lag behind their school peers at the beginning of junior high [39]. In the long run, the high intensity of PSE participation may increase students’ workload, and result in long hours of studying after school and increased stress in competing with peers. Second, PSE dropouts participated in PSE at the beginning of the 7th grade but became less likely to attend PSE from the 7th to 8th grade. By the end of the 8th grade, they no longer participated in PSE, which may indicate their frustration or triumph in PSE participation in terms of academic competition. Based on our findings depicted in the descriptive table, this group of students are more inclined to withdraw from PSE because of frustration. Despite our efforts to capitalize on PSE participation’s heterogeneous patterns, we do not have conclusive evidence or students’ interview data to back up our interpretations. Further research is needed to examine students’ motivation or parent decision making in using PSE to support children’s success in the school achievement.

Our findings suggest that late adopters also risk the trade-off between school achievement and depression symptoms if all students were randomly assigned to participate in PSE. Our results extend previous work on PSE’s effect on adolescents’ school achievement and mental health by demonstrating different PSE participation patterns on the improvement in entrance examination results but also highlight the trade-off students’ mental health.

It is worth noting that nearly 25% of junior high school students (*N* = 1888) did not participate in any PSE activities. These students tend to have the lowest entrance examination scores, lower family socioeconomic status, and lower depressive symptoms in 11th grade. The present study’s findings imply that certain low-income or students living in non-intact families could improve their academic performance if they have chances to gain access to extra learning resources, similar to those offered by PSE, without additional financial burden. While PSE may serve as an extra learning opportunity for students, our findings also show that PSE may not benefit everyone. It could also be a potential risk factor for adolescents’ mental health. As more young people and parents would like their children to go to a prestigious high school or university, it is not easy to ask them to give up trying to use PSE as a means for such a goal. However, identifying the most vulnerable students in terms of the effect of PSE participation on mental health is important, so that students and their parents can develop a practical plan and make an effective decision about the amount of time to participate in PSE per week, when to start PSE, or utilizing stress-free supports to reduce the negative mental health effect of participating in PSE.

This study has several limitations. First, it relies on students’ self-reported measures, such as high school entrance examination scores, 9th-grade academic ranking, or subject-specific efficacy in junior high. Although the correlation between official scores and self-reported school achievement was high, self-reported measures tend to be overstated or understated and could bias our estimates in our attempted causal analysis. More research is called for to use national standardized test scores as students’ outcomes after PSE participation. Using national standardized test scores can minimize reporter bias and reduce the risk of an inflated association between self-reported scores and school achievement. Secondly, the measure of depression symptoms is not a typical instrument for depression, for example, in contrast to the Center for Epidemiological Studies Depression Scale (CES-D). This could present a challenge in the validity of this measurement as an indicator of 11th graders’ mental health. Future research will need to consider a more extensive and systematic measurement of mental health, which could further differentiate PSE’s impacts on different dimensions of mental health. Thirdly, we do not have enough information to identify in detail, the effect of a specific academic subject taken in cram school. Our findings possibly mainly reflect the impact of cram schooling on English and Math, the two most frequently taken subjects at cram schools by junior high students in Taiwan [5]. Since we focus on the impact of various long-term PSE patterns on overall academic achievement and mental health, our findings could be interpreted as the average effects of these PSE patterns across cram-schooled subjects. Of course, it would be better if later studies could show how long-term PSE involvement in a specific subject impacts this subject’s academic achievement. However, how the PSE involvement of a particular subject affects mental health would be a more challenging question to answer. Lastly, we do not have specific measures of school achievement and depression symptoms (e.g., grade point average, CES-D score) prior to the entrance examination or before high school enrollment. Students’ early achievement could affect parents’ and students’ decisions in attending different PSE patterns and their motivation to enroll in PSE. The inclusion of students’ earlier school achievement or depression symptoms may increase or maximize the strength of our findings. Future research, with appropriate data, can look at the long-term trajectory of outcomes or the time-lagged reciprocal effect between school achievement and depression symptoms.

## 5. Conclusions

Given the strong emphasis on school achievement in Taiwanese society, this paper provides an insight into whether there is a trade-off between mental health and school achievement for secondary education students. We identify potential benefits in school achievement and risks in mental health for adolescents who experienced various patterns of PSE participation. This particular study finds that students who are PSE always-takers and dropouts may experience some trade-offs between their senior-high entrance exam scores and depression symptoms. The same trade-off did not happen for PSE early adopters, late adopters, or explorers. In short, the effects of PSE are heterogeneous. PSE may offer some students educational opportunities. It could also be a mental health risk to others.

Increasing PSE participation and depression symptoms in adolescence are becoming global phenomena. Recognizing the negative impact of PSE participation on students’ mental health calls for a thoughtful support system and intervention to prevent academic competency and affects on adolescents’ later mental health and educational attainment [11,27,28]. The findings of the current study provide insight into the arguments for early prevention, school- and home-based intervention for the trade-off dilemma between school achievement and mental health, and the need for large scale integration between prevention and educational policies. We certainly do not think the findings of PSE’s effects in Taiwan could readily be generalized to other societies. With the growing prevalence of shadow education outside of East Asia, it is crucial to know more about how private supplementary schooling beyond the school walls affects adolescents’ health worldwide.

## Figures and Tables

**Figure 1 ijerph-18-01222-f001:**
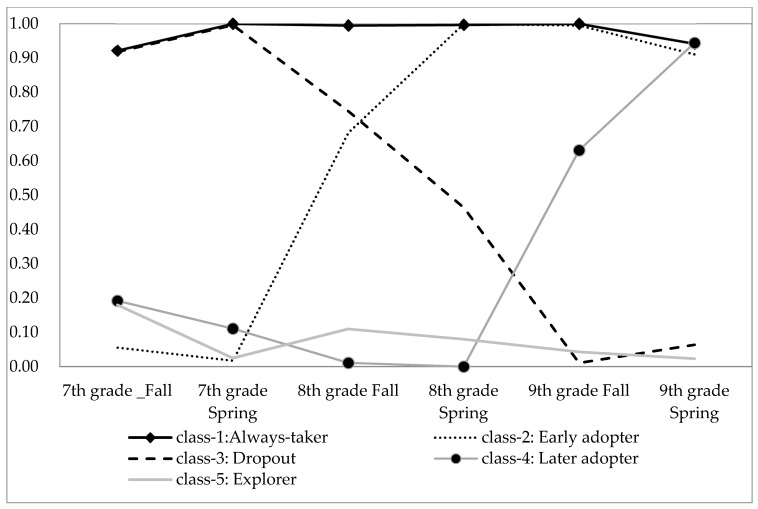
Estimated probability of participating in private supplementary education (PSE) across semesters.

**Figure 2 ijerph-18-01222-f002:**
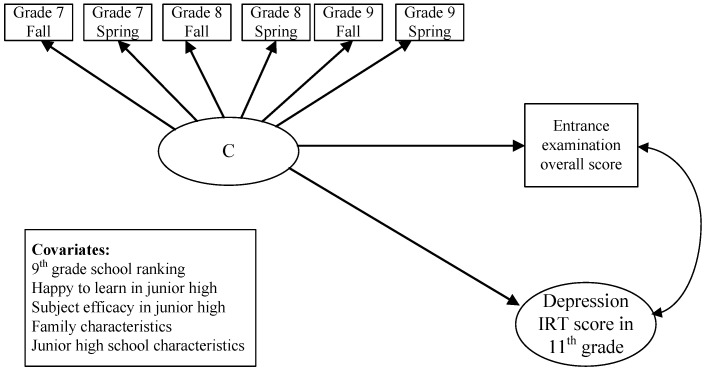
The Research Framework.

**Table 1 ijerph-18-01222-t001:** Descriptive statistics across the patterns of PSE participation.

	Never-Taker	Always-Taker	Early Adopter	Dropout	Later Adopter	Explorer
All variabels ^1^	Mean/(SD) ^3^	Mean/(SD)	Mean/(SD)	Mean/(SD)	Mean/(SD)	Mean/(SD)
Entrance-exam score	6.59	13.85 ***^,2^	13.30 ***	9.41 ***	13.89 ***	10.66 ***
	(6.79)	(7.17)	(6.92)	(7.22)	(7.75)	(8.20)
Depression in 11th grade	−0.03	0.03 *	0.04	0.05	0.07	0.04
	(0.95)	(0.82)	(0.77)	(0.86)	(0.81)	(0.73)
Subject efficacy in junior high	11.51	13.03 ***	12.99 ***	12.22 ***	12.86 ***	12.22 ***
	(2.68)	(2.17)	(2.06)	(2.28)	(2.34)	(2.70)
9th grade academic ranking	3.05	4.08 ***	3.98 ***	3.46 ***	3.96 ***	3.57 ***
	(1.28)	(0.95)	(0.99)	(1.15)	(1.01)	(1.28)
Happiness in learning	2.98	2.94	2.87 **	2.98	2.92	2.95
	(0.76)	(0.74)	(0.72)	(0.70)	(0.75)	(0.77)
Number of siblings	1.50	1.24 ***	1.26 ***	1.32 ***	1.24 ***	1.35 ***
	(1.02)	(0.80)	(0.81)	(0.88)	(0.84)	(0.89)
School climate	0.09	−0.02 ***	−0.02 ***	0.00 **	−0.04 **	0.02 *
	(0.73)	(0.76)	(0.73)	(0.74)	(0.77)	(0.79)
Teacher quality	0.10	0.09	0.01 *	0.04	0.14	0.08
	(0.89)	(0.83)	(0.83)	(0.88)	(0.86)	(0.90)
Family income	3.13	3.48 ***	3.43 ***	3.34 ***	3.44 ***	3.31 ***
	(0.74)	(0.62)	(0.70)	(0.70)	(0.68)	(0.74)
Female	0.41	0.49 ***	0.48 **	0.49 **	0.48 *	0.44
**Ethnicity**						
Minnan/Mainlander	0.91	0.97 ***	0.96 ***	0.95 **	0.95 *	0.93
Aborigines	0.07	0.01 ***	0.01 ***	0.03 ***	0.03 **	0.04 **
Others	0.02	0.01 **	0.02	0.02	0.01	0.02
**Parents’ occupation**						
Farmer	0.06	0.03 ***	0.02 ***	0.04	0.01 ***	0.03 **
Technical	0.25	0.13 ***	0.11 ***	0.17 ***	0.12 ***	0.14 ***
Salary manner	0.26	0.21 ***	0.17 ***	0.25	0.20 *	0.24
Semi-professional	0.23	0.32 ***	0.34 ***	0.31 ***	0.29 *	0.26
Professional	0.20	0.31 ***	0.35 ***	0.23	0.37 ***	0.31 ***
Others: Military	0.01	0.01	0.00	0.00	0.00	0.01
**Parents’ highest education**						
Junior high or lower	0.14	0.04 ***	0.06 ***	0.08 ***	0.02 ***	0.07 ***
High school	0.49	0.37 ***	0.33 ***	0.48	0.27 ***	0.39 ***
Some college	0.24	0.31 ***	0.29 *	0.27	0.31 *	0.27
College	0.09	0.15 ***	0.17 ***	0.12 *	0.19 ***	0.16 ***
Beyond college	0.05	0.12 ***	0.15 ***	0.06	0.21 ***	0.12 ***
**Family structure**						
Intact family	0.70	0.82 ***	0.78 ***	0.71	0.81 ***	0.70
Step-family	0.02	0.01 **	0.01	0.03	0.01	0.01
Single family	0.17	0.10 ***	0.13 *	0.15	0.10 **	0.16
Grand-parent guardians	0.03	0.01 ***	0.02	0.03	0.01 *	0.02
Forster-parent	0.01	0.00 ***	0.00 *	0.00	0.01	0.01
Living with other relatives	0.06	0.05	0.06	0.08	0.06	0.09 **

^1^ All variables except institutional variables are from 2014 survey data collection. ^2^ Two-tailed *t*-tests and Pearson chi-square tests were conducted for continuous and categorical variables and ***, **, * denoted significant differences compared with never-takers under *p* < 0.001 *p* < 0.01, *p* < 0.05. ^3^ Numbers in parentheses are standard deviations.

**Table 2 ijerph-18-01222-t002:** Logistic regression predicting the propensity of PSE participation patterns in junior high.

	Always-Taker	Early Adopter	Dropout	Later Adopter	Explorer
	b/se	b/se	b/se	b/se	b/se
Female	0.458 ***^,1,2^	0.401 ***	0.435 ***	0.361 **	0.189 *
	(0.065)	(0.107)	(0.104)	(0.134)	(0.094)
**Ethnicity (Ref. Minnan/Mainlander)**					
Aborigines	−1.418 ***	−1.273 **	−0.891 **	−0.299	−0.210
	(0.188)	(0.401)	(0.302)	(0.352)	(0.217)
Others	−0.553	−0.352	0.037	−0.588	−0.296
	(0.318)	(0.474)	(0.449)	(0.680)	(0.430)
**Family income**	0.575 ***	0.388 ***	0.397 ***	0.305 **	0.242 ***
	(0.050)	(0.082)	(0.079)	(0.103)	(0.069)
**Parents’ Highest Education**					
High school (Ref. Junior high or lower)	0.784 ***	0.326	0.380	1.331 **	0.484 *
	(0.127)	(0.226)	(0.196)	(0.469)	(0.190)
Some college	1.051 ***	0.698 **	0.402	1.989 ***	0.666 ***
	(0.135)	(0.233)	(0.211)	(0.472)	(0.201)
College	1.125 ***	0.934 ***	0.441	2.310 ***	0.967 ***
	(0.157)	(0.257)	(0.246)	(0.488)	(0.224)
Beyond college	1.354 ***	1.260 ***	0.312	2.972 ***	1.223 ***
	(0.179)	(0.278)	(0.297)	(0.499)	(0.249)
**Family structure** (Ref. Intact family)					
Step-family	−0.921 ***	−0.553	0.311	−0.972	−0.702
	(0.259)	(0.435)	(0.323)	(0.621)	(0.402)
Single family	−0.623 ***	−0.200	−0.148	−0.706 **	0.046
	(0.107)	(0.171)	(0.169)	(0.252)	(0.144)
Grand-parent guardians	−0.944 ***	−0.856 *	−0.120	−1.573 *	−0.288
	(0.221)	(0.426)	(0.316)	(0.738)	(0.312)
Forster-parent	−0.737	−1.462	−0.555	−0.169	0.042
	(0.446)	(1.083)	(0.783)	(0.817)	(0.530)
Living with other relatives or missing flag	−0.408 **	−0.211	0.280	−0.361	0.287
	(0.137)	(0.222)	(0.200)	(0.275)	(0.176)
**Parent Occupation** (Ref. Farmer)					
Technical	0.275	0.066	−0.101	−0.410	0.012
	(0.220)	(0.436)	(0.331)	(0.472)	(0.330)
Salary manner	0.372	0.382	−0.084	−0.196	0.160
	(0.195)	(0.377)	(0.286)	(0.385)	(0.285)
Semi-professional	0.775 ***	0.735	0.193	0.362	0.540
	(0.201)	(0.383)	(0.296)	(0.387)	(0.292)
Professional	0.973 ***	1.181 **	0.294	0.154	0.689 *
	(0.199)	(0.377)	(0.294)	(0.387)	(0.289)
Others: Military	0.834 ***	1.068 **	0.037	0.314	0.668 *
	(0.205)	(0.382)	(0.306)	(0.386)	(0.294)
Number of siblings	−0.274 ***	−0.175 **	−0.113 *	−0.216 **	−0.079
	(0.036)	(0.062)	(0.057)	(0.078)	(0.051)
School climate for learning	−0.261 ***	−0.186 *	−0.165 *	−0.269 **	−0.105
	(0.046)	(0.077)	(0.075)	(0.093)	(0.066)
Teacher quality	0.066	−0.086	−0.026	0.124	−0.007
	(0.040)	(0.066)	(0.063)	(0.081)	(0.056)
Constant	−2.416 ***	−3.589 ***	−2.943 ***	−4.331 ***	−2.690 ***
	(0.266)	(0.486)	(0.402)	(0.650)	(0.379)

^1^ We report regression coefficients. ^2^ *** *p* < 0.001, ** *p* < 0.01, * *p* < 0.05.

**Table 3 ijerph-18-01222-t003:** Estimation of average treatment effect on the treated (ATT) using inverse-probability-of-treatment weighting (IPTW) weighted seemingly unrelated regression (SUR) models by PSE Participation.

	Always-Taker		Early-Adopter		Dropout		Late-Adopter		Explorer	
	Depression	Entrance Exam	Depression	Entrance Exam	Depression	Entrance Exam	Depression	Entrance Exam	Depression	Entrance Exam
	b/se	b/se	b/se	b/se	b/se	b/se	b/se	b/se	b/se	b/se
Always-taker vs. Never-taker ^1^	0.067 *^,2^	1.942 ***								
	(0.032)	(0.220)								
Early-adopter vs. Never-taker			0.048	1.590 ***						
			(0.045)	(0.305)						
Dropout vs. Never-taker					0.089 *	0.767 **				
					(0.045)	(0.268)				
Late-adopter vs. Never-taker							0.067	1.716 ***		
							(0.055)	(0.398)		
Explorer vs. Never-taker									0.048	1.223 ***
									(0.036)	(0.261)
Subjects efficacy in junior high	−0.039 ***	0.406 ***	−0.051 ***	0.475 ***	−0.031 **	0.515 ***	−0.042 **	0.431 ***	−0.038 ***	0.346 ***
	(0.008)	(0.069)	(0.011)	(0.092)	(0.011)	(0.073)	(0.013)	(0.121)	(0.009)	(0.075)
Happiness in learning	−0.159 ***		−0.153 ***		−0.161 ***		−0.182 ***		−0.112 ***	
	(0.025)		(0.037)		(0.037)		(0.042)		(0.031)	
9th grade academic ranking	0.026	3.223 ***	0.054 *	3.144 ***	−0.006	2.850 ***	0.011	3.322 ***	0.025	3.227 ***
	(0.017)	(0.134)	(0.023)	(0.167)	(0.022)	(0.145)	(0.028)	(0.222)	(0.019)	(0.143)
Female	0.036	0.108	−0.022	−0.544	0.048	0.415	0.099	0.233	0.030	0.030
	(0.031)	(0.198)	(0.043)	(0.285)	(0.045)	(0.278)	(0.054)	(0.394)	(0.037)	(0.266)
Aborigines	−0.089	−2.538 ***	−0.049	−3.688 ***	−0.185	−1.296	−0.252 *	−3.000 ***	−0.062	−2.648 ***
	(0.086)	(0.438)	(0.134)	(0.815)	(0.117)	(0.857)	(0.111)	(0.629)	(0.081)	(0.592)
Others	−0.133	3.014	−0.048	2.508	−0.006	−0.027	−0.095	5.989 **	−0.108	0.210
	(0.191)	(2.354)	(0.233)	(1.816)	(0.203)	(1.181)	(0.244)	(2.179)	(0.216)	(1.570)
Family income	−0.083 **	0.003	−0.089 **	0.037	−0.088 *	−0.009	−0.043	0.546	−0.076 **	0.073
	(0.025)	(0.160)	(0.034)	(0.228)	(0.035)	(0.194)	(0.042)	(0.280)	(0.027)	(0.202)
Parents’ highest education										
High school (Ref. Junior high or lower)	−0.043	0.250	0.025	0.094	−0.277 **	0.781	−0.094	−1.264	−0.038	−0.423
	(0.059)	(0.320)	(0.088)	(0.491)	(0.087)	(0.470)	(0.170)	(2.727)	(0.080)	(0.477)
Some college	0.029	2.325 ***	0.065	2.289 ***	−0.150	3.014 ***	−0.180	0.573	0.021	1.686 **
	(0.062)	(0.352)	(0.089)	(0.538)	(0.093)	(0.529)	(0.171)	(2.750)	(0.085)	(0.519)
Normal University	0.154 *	3.579 ***	0.171	3.058 ***	−0.054	3.237 ***	−0.033	1.715	0.158+	2.992 ***
	(0.071)	(0.421)	(0.101)	(0.613)	(0.113)	(0.601)	(0.180)	(2.781)	(0.094)	(0.610)
Beyond college	0.028	5.499 ***	0.108	5.510 ***	−0.038	6.188 ***	0.057	3.445	0.063	5.450 ***
	(0.084)	(0.514)	(0.115)	(0.693)	(0.140)	(0.846)	(0.186)	(2.827)	(0.107)	(0.732)
**Family structure**										
Step-family family)	0.215	−2.492 ***	0.086	−0.030	0.042	−2.080 **	−0.272	−2.745	−0.009	−1.720 *
(Ref. Intact family)	(0.112)	(0.672)	(0.104)	(1.524)	(0.150)	(0.736)	(0.184)	(2.310)	(0.154)	(0.848)
Single family	0.063	−1.104 ***	−0.051	−1.740 ***	0.034	−1.210 **	0.127	−1.779 *	0.014	−0.805
	(0.048)	(0.333)	(0.064)	(0.470)	(0.078)	(0.414)	(0.091)	(0.776)	(0.053)	(0.419)
Grand-parent guardians	−0.204 *	−1.958 ***	0.177	−3.261 ***	−0.138	−1.685 *	−0.366	−1.990	0.060	−2.682 ***
	(0.101)	(0.562)	(0.182)	(0.790)	(0.141)	(0.665)	(0.496)	(2.692)	(0.146)	(0.705)
Forster-parent	0.390 *	−1.411	−0.365	−1.305	−0.017	2.588	0.346	−2.756	0.596 *	−0.657
	(0.169)	(1.147)	(0.522)	(0.961)	(0.279)	(1.788)	(0.225)	(1.487)	(0.280)	(0.852)
Living with other relatives	0.083	−0.755	0.100	−1.601 *	0.116	−0.583	−0.040	−0.877	0.060	0.032
Or missing	(0.063)	(0.440)	(0.079)	(0.624)	(0.079)	(0.551)	(0.100)	(0.873)	(0.073)	(0.569)
**Parent Occupation**										
Technical	0.074	−1.205 *	−0.268	−1.520	0.006	−0.969	−0.215	−1.134	0.012	−1.503
	(0.094)	(0.581)	(0.162)	(1.218)	(0.131)	(0.734)	(0.212)	(1.273)	(0.104)	(0.835)
Salary manner	0.088	−0.175	−0.207	−0.598	0.166	0.177	−0.144	−0.081	0.077	0.035
	(0.083)	(0.511)	(0.143)	(1.051)	(0.095)	(0.530)	(0.192)	(1.014)	(0.074)	(0.728)
Semi-professional	0.170 *	0.463	−0.163	0.006	0.135	0.058	−0.176	0.345	0.172 *	0.709
	(0.085)	(0.533)	(0.144)	(1.079)	(0.101)	(0.591)	(0.193)	(1.047)	(0.079)	(0.763)
Professional	0.118	0.093	−0.210	−0.918	0.205 *	0.477	−0.181	0.178	0.105	0.309
	(0.085)	(0.530)	(0.144)	(1.059)	(0.098)	(0.582)	(0.192)	(1.052)	(0.078)	(0.757)
Others: Military	0.162	1.065	−0.119	0.229	0.183	1.019	−0.154	1.309	0.191 *	1.333
	(0.087)	(0.549)	(0.146)	(1.079)	(0.104)	(0.594)	(0.193)	(1.012)	(0.083)	(0.779)
Number of siblings	−0.001	−0.662 ***	0.025	−0.894 ***	−0.018	−0.349 *	0.012	−0.819 ***	−0.012	−0.360**
	(0.017)	(0.123)	(0.025)	(0.168)	(0.022)	(0.138)	(0.031)	(0.242)	(0.020)	(0.133)
School climate	−0.085 ***	−0.236	−0.086 *	−0.383	−0.116 **	−0.797 ***	−0.081	−0.534	−0.085 **	−0.386
	(0.025)	(0.177)	(0.038)	(0.252)	(0.036)	(0.217)	(0.042)	(0.310)	(0.030)	(0.214)
Teacher quality	−0.001	−0.020	−0.014	−0.191	−0.021	0.212	−0.013	0.082	0.009	0.007
	(0.020)	(0.127)	(0.027)	(0.177)	(0.030)	(0.165)	(0.034)	(0.246)	(0.024)	(0.166)
Constant	0.956 ***	−7.873 ***	1.267 ***	−6.957 ***	1.150 ***	−9.071 ***	1.355 ***	−8.523 **	0.791 ***	−7.544 ***
	(0.161)	(0.954)	(0.237)	(1.573)	(0.207)	(1.121)	(0.317)	(2.850)	(0.173)	(1.228)
$\sigma_{11}$ (Depression)	0.718 ***		0.676 ***		0.756 ***		0.682 ***		0.660 ***	
	(0.019)		(0.026)		(0.029)		(0.033)		(0.025)	
$\sigma_{22}$ (Entrance exam)	28.798 ***		26.470 ***		25.999 ***		31.494 ***		28.597 ***	
	(0.869)		(1.135)		(1.075)		(1.634)		(1.209)	
$\sigma_{12}\;$(Depression*Entrance exam)	0.253 **		0.208+		0.407 ***		0.278+		0.262 **	
	(0.084)		(0.112)		(0.108)		(0.148)		(0.102)	
$\sigma_{12}$ correlation residual	0.055 *		0.049		0.091 *		0.0601		0.060 *	

^1^ We run the SUR model by PSE participation patterns. All SUR models compared to students who never participate in PSE (Never-taker). ^2^ *** *p* < 0.001, ** *p* < 0.01, * *p* < 0.05.

**Table 4 ijerph-18-01222-t004:** Estimation of average treatment effect (ATE) using IPTW weighted SUR models by PSE participation.

	Always-Taker ^1^		Early-Adopter		Dropout		Late-Adopter		Explorer	
	Depression	Entrance Exam	Depression	Entrance Exam	Depression	Entrance Exam	Depression	Entrance Exam	Depression	Entrance Exam
	b/se	b/se	b/se	b/se	b/se	b/se	b/se	b/se	b/se	b/se
Always-taker	0.068 *^,2^	2.071 ***								
vs. Never-taker	(0.031)	(0.205)								
Early-adopter			0.058	1.922 ***						
vs. Never-taker			(0.048)	(0.284)						
Dropout					0.104 *	0.891 **				
vs. Never-taker					(0.047)	(0.272)				
Late-adopter							0.139 *	2.238 ***		
vs. Never-taker							(0.062)	(0.489)		
Explorer									0.051	1.164 ***
vs. Never-taker									(0.036)	(0.233)
$\sigma_{11}$	0.735 ***		0.690 ***		0.766 ***		0.708 ***		0.674 ***	
	(0.018)		(0.026)		(0.030)		(0.036)		(0.026)	
$\sigma_{22}$	28.211 ***		23.550 ***		25.064 ***		30.744 ***		25.867 ***	
	(0.819)		(0.940)		(1.068)		(2.747)		(1.044)	
$\sigma_{12}$	0.252 **		0.229 *		0.430 ***		0.015		0.326 ***	
	(0.079)		(0.100)		(0.104)		(0.198)		(0.096)	
$\sigma_{12}\;$	0.055 *		0.056 *		0.098 *		0.003		0.078 *	

^1^ We run the SUR model by PSE participation patterns. All SUR models compared to students who never participate in PSE (Never-taker). ^2^ *** *p* < 0.001, ** *p* < 0.01, * *p* < 0.05.

## Data Availability

Restrictions apply to the availability of these data. Data was obtained from Taiwan Upper Secondary Education Database (TUSED) and are available from https://use-database.cher.ntnu.edu.tw/used/ with the permission of TUSED.

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
