# Peer review of "The Heterogeneous Effects of Participation in Shadow Education on Mental Health of High School Students in Taiwan"

_ijerph, 2021, doi:10.3390/ijerph18031222_

Round 1
Reviewer 1 Report
Authors managed to focus on the various patterns of PSE participation and on the impacts of these patterns on the outcomes of school achievement and depression.
(1). In this study authors examined many variables, but the subject is very complex and cannot be restricted.
(2). So, the findings from the survey do not lead to concrete conclusions.
(3). It is interesting (and I think it is expected) that the always-takers have significantly higher depression symptoms in high school than the never-takers!
(4). Results and discussion can trigger more research in the future.
Reviewer 2 Report
The study on this topic is very interesting. The structure is clearly and logical and challenging.
The main goal of the study was analyzing the heterogeneous effect of private supplementary education (PSE) participation on school achievement and depression symptoms among high schoolers.
The research design and process seem to be adequate to the purpose and are clearly described in the text.
I think that the quality of the results discussion and conclusions could be improved. In the discussion section will be interesting bring to the discussion other studies and authors to support the findings.
What links can be established with previous studies?
How can the results be read considering the relevant literature on this topic?
Do the results corroborate or contradict previous studies?
The conclusions of the study need to be extended, pointed out the main conclusions and suggesting future research line to mitigate the methodological and context limitations.
What are the main conclusions of the study and what do they bring to this field again?
What are the main conclusions of the study and what is new for this field of research?
In the text, some limitations were pointed out by the authors in the discussion section. Taking these limitations in account what kind of link could be made with new research studies?
